# Patient perceptions of vulnerability to recurrent respiratory tract infections and prevention strategies: a qualitative study

Laura Dennison ![ORCID],[1] Sian Williamson ![ORCID],[1] Kate Greenwell ![ORCID],[1] Molly Handcock,[2] Katherine Bradbury,[1,3] Jane Vennik ![ORCID],[2] Lucy Yardley,[1,4] Paul Little ![ORCID],[2] Adam W A Geraghty ![ORCID][2]

[1]Centre for Clinical and Community Applications of Health Psychology, School of Psychology, Faculty of Environmental and Life Sciences, University of Southampton, Southampton, UK
[2]School of Primary Care, Population Sciences and Medical Education, Faculty of Medicine, University of Southampton, Southampton, UK
[3]NIHR Applied Research Collaboration (ARC) Wessex, Southampton, UK
[4]School of Psychological Science, University of Bristol, Bristol, UK

**Correspondence to**
Dr Laura Dennison;
l.k.dennison@soton.ac.uk

## ABSTRACT

**Objectives** Respiratory tract infections (RTIs) are extremely common, usually self-limiting, but responsible for considerable work sickness absence, reduced quality of life, inappropriate antibiotic prescribing and healthcare costs. Patients who experience recurrent RTIs and those with certain comorbid conditions have higher personal impact and healthcare costs and may be more likely to suffer disease exacerbations, hospitalisation and death. We explored how these patients experience and perceive their RTIs to understand how best to engage them in prevention behaviours.

**Design** A qualitative interview study.

**Setting** Primary care, UK.

**Methods** 23 participants who reported recurrent RTIs and/or had relevant comorbid health conditions were interviewed about their experiences of RTIs. Interviews took place as the COVID-19 pandemic began. Data were analysed using inductive thematic analysis.

**Results** Three themes were developed: Understanding causes and vulnerability, Attempting to prevent RTIs, Uncertainty and ambivalence about prevention, along with an overarching theme; Changing experiences because of COVID-19. Participants' understandings of their susceptibility to RTIs were multifactorial and included both transmission via others and personal vulnerabilities. They engaged in various approaches to try to prevent infections or alter their progression yet perceived they had limited personal control. The COVID-19 pandemic had improved their understanding of transmission, heightened their concern and motivation to avoid RTIs and extended their repertoire of protective behaviours.

**Conclusions** Patients who experience frequent or severe RTIs are likely to welcome and benefit from advice and support regarding RTI prevention. To engage people effectively, those developing interventions or delivering health services must consider their beliefs and concerns about susceptibility and prevention.

## INTRODUCTION

Respiratory tract infections (RTIs) are infectious diseases of the upper or lower respiratory tract that are usually caused by viruses. RTIs include the common cold, influenza, pharyngitis, tonsillitis, sinusitis and bronchitis along with pandemic infections such as COVID-19.

## Strengths and limitations of this study

► This study contributes to a limited evidence-base on the experiences of people who suffer recurrent respiratory tract infections (RTIs) and/or are at risk of complications, morbidity and mortality as a consequence of RTIs.
► The interview technique and inductive thematic analysis allowed an in-depth exploration of beliefs and experiences, prioritising the participants' perspectives.
► The interview timing coincided with the beginning of the COVID-19 pandemic, allowing interesting observations of how experiencing the pandemic and its restrictions affected people's understandings and behaviour relating to non-pandemic RTIs.
► The purposive sampling strategy ensured that participants had a range of clinical and demographic characteristics but may have failed to capture underserved populations; particularly minority ethnic groups and people with high levels of deprivation.

RTIs are common with most adults experiencing at least one cold per year.[1] RTIs typically produce mild to moderate symptoms and are usually self-limiting. Nonetheless they are a leading cause of work sickness absence[2] and reduce health-related quality of life, affecting physical, mental and social functioning.[3] They are also responsible for substantial healthcare costs, mostly in primary care.[4 5] Consultations for RTIs can also result in unnecessary antibiotics prescribing, contributing to antimicrobial resistance.[6 7] Certain subgroups of patients have worse health outcomes and incur higher healthcare costs from RTIs. This includes patients who experience frequent or recurrent infections and those whose older age and/or comorbid conditions put them at higher risk of disease exacerbations, hospitalisation and death.[8 9] RTI prevention interventions, particularly those focused on these recurrent and at-risk patients, would benefit individuals and healthcare systems alike.

A wide range of medical and behavioural RTI prevention approaches have been investigated including; hand hygiene,[10–12] vaccination,[13–15] social distancing,[10 16] mask-wearing,[10 11 17] vitamins and supplements,[18 19] gargling, nasal rinses and nasal sprays,[20] exercise[21 22] and mindfulness meditation.[21] Interest and urgency around RTI prevention has been prompted by the COVID-19 pandemic. However, in addition to efficacy, prevention interventions must be acceptable to the target population; they must feel personally relevant, feasible and make sense in the context of people's experiences. Therefore, gaining insight into how people understand and experience RTIs and RTI prevention is a vital step in the development and implementation of successful prevention interventions.[23]

Many studies have investigated lay perspectives on RTIs, using either qualitative designs or survey methods. Insights that include colds and influenza are considered familiar and trivial,[24–26] with people often considering themselves to be at low risk of both contagion and severe outcomes compared with others.[25 27–29] Transmission from symptomatic people is often correctly recognised as important.[24 26 29 30] Confusion may exist about the importance of viruses versus bacteria and the relevance of temperature, damp and pollution.[25 30–35] RTI severity seems to be judged based on symptom duration, discomfort and impact on activities.[25 36 37] People describe attempting to manage RTIs at home.[25 32 36 37] Seeking a General Practice (GP) consultation tends to be triggered by long symptom duration, specific symptoms[25 38] or where symptom resolution in earlier illness episodes has been attributed to antibiotics.[36 37] Considerable fatalism exists about catching RTIs.[24 26 39 40] Furthermore, while people may be familiar with and adopt prevention approaches such as avoiding infectious people and hand-washing,[35 41] these behaviours can be considered unacceptable and/or unattainable.[27 29 40] People tend to emphasise their own good health, strong personal reserves and healthy lifestyle as providing defence against RTIs.[24 27 30 41] Experiencing RTIs may even be considered beneficial: a natural, necessary strengthening of the immune system.[24 26]

There are some important limitations to the scope of existing research. First, most studies sample the general public, rather than patients with higher infection frequency, morbidity or mortality. Second, most studies focus specifically on colds and influenza rather than the wider range of RTIs. Furthermore, this literature is dominated by understanding patient perspectives of RTIs specifically in the context of reducing antibiotic prescribing and increasing influenza vaccine uptake.

The current paper extends the literature by focusing on patients who get recurrent RTIs and/or are at risk for having more severe consequences of RTIs. We explored how these patients experienced and understood their infections in order to understand how best to engage them in prevention behaviours. This research was planned prior to the COVID-19 pandemic and was nested within an National Institute for Health Research programme grant developing and evaluating RTI prevention approaches. This work was originally driven by the impact of recurrent non-pandemic RTIs on patients and healthcare systems. The pandemic, and the resulting requirement for both 'at risk' patients and whole populations to adopt and adhere to an array of RTI prevention behaviours, highlights and accelerates the need to understand people's experiences of and beliefs about RTIs, their vulnerability and infection prevention behaviours.

## METHOD

We report this qualitative interview study with reference to the Consolidated Criteria for Reporting Qualitative Research(COREQ) checklist (online supplemental material 1).

### Recruitment

We recruited adults (18+) who had experienced recurrent RTIs (operationalised as ≥3 in the last year and/or ≥1/year over the last 3 years). Because RTIs are common and usually self-limiting, they are not routinely recorded in medical records (only when patients consult). Therefore, a combination of medical record searching and self-report was used to identify participants. Three Hampshire (UK) GP practices identified possible participants. The practices were chosen to capture a mixture of urban and semi-rural locations and varying deprivation levels. Practices searched their lists and posted invitations and information sheets to (a) patients who consulted for ≥1 RTI within the last year and received an antibiotics prescription and (b) a subgroup of patients who also had asthma, Chronic Obstructive Pulmonary Disease (COPD) or chronic sinusitis. Interested patients returned reply slips, on which they self-reported their recent RTI history. We then purposively sampled from the 52 interested patients to achieve a heterogeneous sample based on frequency of self-reported RTIs, health conditions, age and gender. Twenty-three participants were interviewed. Participants received a £10 retail voucher to thank them for their time. Data collection and analysis were conducted in parallel. Recruitment ceased when patients with a range of clinical and demographic characteristics had been sampled and sufficient data were available to support both the thematic analysis of RTI experiences and beliefs and the iterative intervention refinement (see below).

### Data collection

Data collection took place between March and August 2020, coinciding with the beginning of the COVID-19 pandemic. Three participants were interviewed face-to-face in their homes or University premises in early March. The rest participated by telephone during the first UK lockdown or as national restrictions eased. After gathering brief demographic and health data, participants were interviewed following a schedule consisting of two sections (see online supplemental material 2).

Section 1 involved an in-depth exploration of their personal experiences of having RTIs, including the types of infections they experience, how infections affect them, when they experience infections, why they believe they experience them, how they manage them and if and how they attempt to avoid them. Later interviews also included questions on the impact of COVID-19 and the pandemic restrictions on their experiences of RTIs. Questions were phrased openly and non-directive prompts were used to elicit extensive discussion.

Section 2 gathered data on participants views and experiences of web-based RTI prevention interventions that were under development. Participants used and discussed either an intervention promoting nasal spray use (n=13) or physical activity and stress management interventions (n=10). Some of the interviews were think-aloud style (n=14), others were post-intervention use interviews (n=7). Two participants engaged in both types of interview, over two different sessions. Section 2 data collection was primarily intended to be used for a iterative intervention optimisation, as part of a wider project (reported elsewhere) where the data were used to ensure that the interventions were engaging, persuasive and relevant.[23] However, as participants viewed and discussed the interventions they tended to elaborate on their personal RTI experiences and beliefs. Therefore, any section 2 data relating to current study research questions were included in the current analysis.

SW and LD conducted the interviews (interviewing 20 and 3 participants, respectively). Both interviewers were female academics, with qualitative interviewing and health psychology expertise. Interviews ranged from 41 to 104 min (mean=70.5), including both sections 1 and 2 . Interviewers took field notes. Interviews were audio-recorded then transcribed verbatim.

## Data analysis
An inductive thematic analysis was conducted from a critical realist perspective.[42] The analysts have health psychology (LD, KG, SW, AWAG), medicine or health-care (MH, JV) research backgrounds and conducted the analysis as a study nested within a programme of research relating to behavioural approaches to RTI prevention. LD conducted the analysis, periodically presenting and discussing preliminary codes and themes and with co-authors SW, KG, JV, AWAG and MH. This practice allowed multiple perspectives to be considered, avoided idiosyncratic interpretations and helped to highlight modifications or clarifications to improve the analysis. The authors do not consider an unbiased or objective qualitative analysis as achievable and instead aim to present descriptive and interpretative accounts of patterns in the data that have been derived from a rigorous engagement with the data using analytic procedures that stay close to the original data in order to derive inductive themes.

To begin the analysis, familiarisation with the data was achieved through listening to audio-recordings and reading transcripts. Next, thorough line-by-line coding began, facilitated by NVivo V.12 (QSR International, Burlington, Massachusetts, USA) software. Descriptive labels (codes) were attached to words or phrases that captured ideas relating to the research questions. Codes were reviewed frequently, and definitions were developed and iteratively edited. The analysis proceeded to examine the codes and develop them into themes which captured patterns and features of the data. This involved an iterative process of clustering together or merging similar codes and splicing or dividing codes, all the time engaging in constant comparison between codes and transcripts and searching for deviant cases. Themes were iteratively reviewed, refined, organised and relabelled, with the input of the wider team, to ensure the final set of themes and subthemes were insightful and coherent.

## Patient and public involvement (PPI)
A panel of five PPI contributors have supported the study planning and conduct, some from the grant application stage. The PPI panel includes people with experiences of recurrent infections and/or long-term health conditions that make them vulnerable to more severe RTIs. Contributions included editing and improving our participant information sheets, consent forms and interview schedules. They also participated in pilot interviews to ensure minimal participant burden and improve the question coverage, wording and process.

## RESULTS
### Participants
Table 1 shows participant characteristics. Participants self-reported a mean of three infections in the previous year. Ten participants had asthma, three had COPD and three had chronic sinusitis. More women than men participated. Age ranged between 18 and 83.

### Themes and subthemes
Three themes and 12 subthemes were developed from the data (table 2). A further, overarching theme which influenced all other themes is described. This final theme relates to the evolving nature of RTI experiences, beliefs and behaviours given the emerging pandemic context. The themes are explained below, alongside illustrative quotations.

### Understanding causes and vulnerability
Participants had complex, multidimensional understandings about why RTIs occur and how and why they are more susceptible than others.

### Catching RTIs from other people
All participants understood that RTIs were transmitted between people, identifying people and situations that they considered pertinent for their own infections. Some discussions included the relevance of symptomatic people, highlighting coughs and sneezes as transmission mechanisms and occasionally drawing on the concepts of germs and viruses.

**Table 1** Demographic and clinical characteristics of participants (n=23)

| | Mean (SD), range or n (%) |
|---|---|
| Age (years) | 53.87 (19.62), range 18–83 |
| Gender | |
| Male | 6 (26.09%) |
| Female | 17 (73.91%) |
| Marital status | |
| Married or living with partner | 11 (47.83%) |
| Single | 5 (21.74%) |
| Divorced | 4 (17.39%) |
| Widowed | 3 (13.04%) |
| Employment status | |
| In paid work (full or part time including self-employed) | 10 (43.48%) |
| Retired | 6 (26.09%) |
| Not working because of illness/disability | 2 (8.70%) |
| Other (unemployed, homemakers, students, volunteers) | 5 (21.74%) |
| Education (age left education) | |
| 16 or before | 3 (13.04%) |
| 17 or 18 | 6 (26.09%) |
| Over 18 | 14 (60.87%) |
| Ethnicity | |
| White British or White Irish | 16 |
| Mixed White/Asian | 1 |
| No data | 6 |
| Deprivation (2019 Index of Multiple Deprivation Decile*) | Mdn=9 (IQR 6.5), range 2–10 |
| Health conditions† | |
| Asthma | 10 (43.48%) |
| COPD | 3 (13.04%) |
| Chronic sinusitis | 3 (13.04%) |
| None of these conditions | 12 (52.17%) |
| Number of RTIs in last 12 months (self-report) | 3.23 (1.6), range 1–6 |
| Number of RTIs per year in last 3 years (self-report) | |
| ≥1 | 21 (91.3%) |
| ≥3 | 15 (65.2%) |
| Types of RTIs experienced at least once in last 12 months (self-report) | |
| Cold | 17 (73.91%) |
| Influenza | 4 (17.39%) |
| Throat infection | 13 (56.52%) |
| Chest infection | 12 (52.17%) |

Continued

**Table 1** Continued

| | Mean (SD), range or n (%) |
|---|---|
| Sinus infection | 10 (43.48%) |
| Ear infection | 4 (17.39%) |
| Concern about RTIs (0–10)‡ | 5.36 (3.09), 0–10 |
| Consequences of RTIs (0–10)‡ | 6.04 (3.08), 1–10 |

*(1=most deprived, 10 is least deprived), data available for 17/23 participants.
†Four participants had more than one of these health conditions.
‡Brief Illness Perception Questionnaire (B-IPQ) concern and consequences items, higher scores indicate higher concern and worse perceived impact.
COPD, Chronic Obstructive Pulmonary Disease; RTI, respiratory tract infection.

If you're next to somebody who's coughing and spluttering, the chances are you'll end up with the same. (P21, female, 60s)

Participants emphasised proximity to others as important for spreading infection and physical contact as particularly risky.

(On the Underground) you're like sardines in a can and if there's anything going you're going to catch it. (P9, male, 80s)

Where people are hugging and kissing. That's when these things get passed around. (P23, female, 50s)

A few participants also described getting infections from contaminated objects and surfaces.

Many participants emphasised children as frequent and significant sources of infection.

Little germ machines. (P23, female, 50s)

Super-spreaders. (P4, female, 60s)

Children were described as having frequent infection and being likely to pass infections on to their families. Their poor respiratory hygiene was highlighted.

Particularly revolting and snotty. (P21, female, 60s)

Children posed particular problems for older participants who were wary of catching RTIs.

(My grandchildren) had a cold and I didn't say anything because they are glad to come down to see us, but I was thinking they were sat on that (chair) and I was thinking 'please don't get any closer'. (P1, male, 60s)

Several participants highlighted their current or past occupation as incurring high RTI risk. They described exposure to large numbers or certain types of people (eg, children, students, general public), or close contact (eg, a tattoo artist), as well as commuting on public transport.

**Table 2** Themes and subthemes

| Overarching theme | Themes | Subthemes |
|---|---|---|
| Changing experiences because of COVID-19 | Understanding causes and vulnerability | Catching RTIs from other people<br>Something in the environment<br>Defective bodies and inadequate defences<br>Wondering why |
| | Attempting to prevent RTIs | Trying (desperately) not to get RTIs<br>Hygiene measures<br>Avoiding symptomatic people<br>Keeping yourself strong and healthy<br>Nipping infections in the bud |
| | Uncertainty and ambivalence about prevention | Inevitability and limited control<br>Judging whether things help<br>Not letting it dominate |

RTI, respiratory tract infection.

### Something in the environment

Many participants referred to environmental influences as responsible for their RTIs. Season was referenced as a very obvious factor by many although a few described having year-round infections.

> Most of the winter months, I am like, 'Oh, I've got a cold again.' (P2, female, 30s)

A few considered exposure to cold air and/or temperature change as causing infections and/or provoking symptoms.

> Maybe when I go out and I don't keep warm enough, so my chest is exposed. (P14, female, <20s)

> Going out into the cold, coming back to the heat, and I think that's just where people get colds from. (P8, female, 50s)

A few participants described mixing more with people indoors as relevant to seasonality and, consistent with their understandings of transmission via people, participants sometimes referred to breathing virus-contaminated air. A few participants perceived pollution, damp or sensitivity to triggers in the atmosphere (eg, deodorants, cleaning products) as potential causes of their RTIs. Those who emphasised poor air quality and environmental triggers tended to also discuss having asthma and/or allergies.

### Defective bodies and inadequate defences

In addition to the causal factors mentioned above, most participants perceived that deficiencies within their bodies had created vulnerabilities to RTIs. Many participants discussed picking up infections more readily than others, experiencing them more severely and finding them harder to shift.

> I just seem to pick up everything. (P4, female, 60s)

> (My husband) doesn't tend to suffer as badly as I do. We joke about it that I make more fuss, but it isn't that. Mine just goes on for longer and is always worse. (P12, female, 70s)

Participants described how ageing and age-related lifestyle influenced RTI vulnerability. A few of the older participants spoke of their advanced age as weakening their ability to fight off infections.

> When you're older you're just not as strong or it's not as easy to shrug these things off. (P1, male, 60s)

However, some older people found that ageing-related lifestyle changes and retirement had reduced exposure to people, making them less susceptible to picking them up.

Several participants felt their chronic health conditions, or a previous severe RTI meant their body did not fight off RTIs effectively, describing weakened defences or immune systems.

> It's just a little bit weaker and just that little kind of chink in its armour is enough to let infections in whereas previously (before asthma) I was a bit healthy and robust and the infection would be fought off instantly by my own immune system. Now, it's just a little bit tired and it just can't cope quite as well. (P18, female, 50s)

> My immune system is rock bottom at the moment where I've just come out the other side of (a lengthy recovery from a severe RTI) so that makes me, in my head, more vulnerable. I think I could easily pick up something else. (P15, female, 50s)

Some discussed how past medical treatments, may have left them with weakened defences. P4 (female, 60s) described radical sinus surgery over 30 years ago that had left her with 'no natural defences at all' and P12 (female, 70s) suspected that her treatment for a thyroid condition had affected her immune system.

Some participants believed a part of their body was weakened, malfunctioning or malformed.

> A slight weakness on the throat. (P3, male, 70s)

> There's something, somewhere in me, that decides it's going to not work properly. (P7, female, 80s)

Something wonky in my nose. (P17, female, 40s)

Participants also discussed the importance of stress. Many participants believed that experiencing upsetting, difficult experiences made people in general more likely to suffer RTIs (and other illnesses) or make your RTI more severe. A few participants considered positive mental health and happiness to be protective, providing a better ability to fight infections. Many also considered stress pertinent to their personal experience of recurrent or severe RTIs.

That's why I succumbed to being so unwell because of the stress that I was under. I think that was a massive, played a massive part. (P15, female, 50s)

A few participants felt that vulnerability tended to occur after stress or busyness, and discussed a perception of succumbing at the point of beginning to relax or take a break. In contrast, some participants did not perceive a role for stress in getting infections. Some emphasised an absence of stress (particularly older participants), citing a lack of life stressors or a lack of personal stress reactivity ('I'm not that sort of person', P17, female, 40s). Some of these participants, however, discussed busyness and exhaustion as personally pertinent; emphasising being 'run down' rather than 'stressed'.

### Wondering why

Despite offering the explanations of personal vulnerability and describing an understanding of viral transmission outlined above, some participants simultaneously referred to not fully understanding why they suffered regularly or severely with RTIs. Some expressed a residual, enduring sense of mystery surrounding their RTIs and a tendency to review and analyse their own illness episodes to try to understand and explain their experiences. The unknowns sometimes related to being unable to trace the infection's origin, for instance, where participants could not identify how or where they had been exposed to RTIs. Some participants struggled with explaining why their own infections were worse than those of people they knew who had seemingly caught the same infection.

There seems to be no rhyme or reason to it because nobody else seems to be ill around me, it just seems to be me. (P7, female, 80s)

Importantly, claims to not understand why they suffered RTIs appeared not only to reflect genuine confusion and uncertainty. They could also function as a conversational device to communicate a sense of injustice; that their infections are undeserved. Unfairness was keenly felt, given their cautious behaviour and attempts to look after their health.

*[Interviewer: Why do you think you get these infections?]* Participant: That's something I ask myself many, many times. I've got friends who are older than me and they don't suffer like I do. I've got friends who are younger than me. They go out drinking. They

smoke. They also indulge in other things. They don't suffer like me. (P22, female, 70s)

### Attempting to prevent RTIs

Almost all participants wanted to prevent RTIs and were taking steps to try to protect themselves.

#### Trying (desperately) not to get RTIs

Some participants' attempts to avoid getting RTIs involved multiple, complex behaviours and regimes.

When you know that you have a tendency to be poorly when you catch things, you just do everything you can to not catch them. (P21, female, 60s)

Many RTI prevention strategies were long-term habits, often learnt in childhood (eg, hand and respiratory hygiene). These behaviours were considered obvious and given little thought.

It's just been second nature. (P15, female, 50s)

Participants also described their repertoire of RTI prevention behaviours as evolving over time, with new products and strategies adopted following recommendations from family, friends and healthcare professionals. Participants were open to and enthusiastic about trying prevention approaches.

You can get really fed up of getting ill and you can be like desperate to latch on to anything. (P13, female, 20s)

#### Hygiene measures

Most participants described hygiene measures to prevent germs getting into their system. Handwashing was almost universally reported, usually as something obvious.

All the normal hygiene, hand-washing that sort of thing (P17, female, 40s)

Some mentioned handwashing very briefly, others described specifics of their meticulous regimes. A few described other hygiene measures including cleaning objects and safe tissue disposal.

If someone comes in and they say they're feeling a bit run down then I'll make sure that I've sterilised and cleaned the room and my hands properly. (P5, female, 20s)

Mask wearing and ventilation were discussed infrequently. A few people mentioned avoiding touching their face as important, but difficult. In general, participants described their exemplary hygiene behaviour but highlighted poor practices of others.

They've sneezed, and I felt it on the back of my head. (P1, male, 60s)

I wish more people did it (handwashing). (P23, female, 50s)

## Avoiding symptomatic people

A key prevention strategy our participants adopted was avoiding symptomatic people. Most described deliberately avoiding anyone displaying or declaring cold symptoms.

I will take a very wide berth. (P10, female, 60s)

I'm just, 'Stay away from me'. (P1, male, 60s)

Some were extremely vigilant about avoiding ill people.

Everybody knew not to come and visit if they had so much as a sniffle. (P23, female, 50s)

However, other participants described how this is impossible or undesirable (see the section 'Not letting it dominate').

## Keeping yourself strong and healthy

Many participants described undertaking health-enhancing behaviours such as eating healthily, taking vitamins, keeping physically fit, nurturing psychological well-being, not smoking, alcohol avoidance and staying hydrated. However, these behaviours were rarely adopted or sustained with preventing RTIs as the end goal. Instead they were long-term habits or preferences or related to healthy ageing or managing other health conditions

That's what I've always done. That is who I am. (P4, female, 60s)

It's good life practice. (P6, male, 70s)

Nonetheless, participants believed or hoped that these healthy behaviours would provide some protection from RTIs.

I would hope that it means that I'm a bit more resilient to fighting them off. (P21, female, 60s)

Unlike the behaviours mentioned above, a few behaviours were adopted specifically for preventing RTIs. Several participants discussed the influenza vaccination (with mixed uptake and views on efficacy and safety). A few ensured they kept warm in cold weather. Several participants with asthma or COPD described 'keeping on top of' their medications, monitoring indicators of health conditions (eg, peak flow) or having a medication escalation regime in place.

It just keeps your body at an even keel, don't it? Because if you let your medication drop, then your body immune system will drop as well. (P8, female, 50s)

## Nipping infections in the bud

Besides attempts to prevent infections, many participants discussed taking prompt action when RTI symptoms began and attempting to alter the infection's course.

When I know [a] cold's coming […] I try and stop it going any further. (P17, female, 40s)

(if) I don't deal with a suspected cold quickly, it will take hold. (P10, female, 60s)

Through experience, participants recognised early infection symptoms and subsequently tried to 'nip infections in the bud', stopping symptoms progressing and worsening. Some found this was at least sometimes effective. The behaviours considered helpful varied and included early use of over-the-counter cold and influenza remedies, extra vitamins or supplements, gargling, nasal rinses and nasal sprays or swift GP consultation for antibiotics.

With a sore throat, sometimes I just go straight home and gargle […] I get the tickle, get the thing, it goes away in a couple of days. (P3, male, 70s)

Obviously, I try and get some antibiotics. If I can get them early, I can stop it getting hold. If it gets hold, there's nothing I can do. (P22, female, 70s)

Participants with asthma and COPD discussed altering medications in response to suspected beginnings of infections.

I do 'up' (increase) my steroid inhaler, so that kind of gives my lungs a boost before the illness strikes and then can usually fend things off that way, but if I don't feel it coming on, then I have to fight it once it arrives, which is slightly harder. (P18, female, 50s)

## Uncertainty and ambivalence about prevention

Most participants discussed the extent to which RTI prevention seems possible or desirable.

### Inevitability and limited control

Despite their concerted prevention efforts, most participants described feeling somewhat powerless over contracting infections, with some discussing the inevitability of symptoms worsening and progressing.

You can do all these things to try and prevent but if you're going to get something you're going to get it I think! (P15, female, 50s)

I've done everything I possibly can to try and prevent it happening and not making it as bad as what it is, but it just never seems to work, and I always end up at the doctor's with steroids, antibiotics. (P11, female, <20)

Central to the perceived inevitability was the understanding that RTIs come from other people. Problems arose because is not always obvious who has infections nor is it always possible or desirable to avoid people. Participants often described feeling at the mercy of other people's behaviour and hygiene.

I've got all my systems are all in place to avoid all those things. It's only really when I step out anywhere else that I'm in an environment, that I can't control (P23, female, 50s)

I can't control everybody else, the snotty noses and the likes. (P2, female, 30s)

### Judging whether things help

As discussed in the previous themes, sometimes participants had experienced perceptible reductions in RTIs as a result of nipping infections in the bud or avoiding people (see previous theme). However, more often participants described uncertainty about whether the strategies they adopted were helpful. Some seemed indifferent to scrutinising helpfulness, lacking curiosity about habitual behaviours. Some spoke of the difficulties of evaluating their effectiveness.

> Whether they do any good or not I have no idea, because I feel exactly the same as I did before. (P13, female, 20s)

> I suppose I'd have to not take them (vitamins) for a year and then try and see the difference. (P10, female, 60s)

Sometimes participants abandoned strategies that did not produce obvious benefits but sometimes they persevered through habit or hope.

> I'm assuming they're doing something! (P19, female, 40s)

> I'm going to continue doing it because it might make a difference soon, or 1 day. (P11, female, <20)

### Not letting it dominate

Participants often discussed how a balance between RTI prevention and quality of life must be achieved. Some RTI avoidance behaviours were considered undesirable or almost impossible, particularly those that involved social distancing from even asymptomatic people.

> You can't put yourself into a bubble and do nothing. That would be a very destructive way of living. Then, you become victimised by being unwell which is no good to anybody. (P4, female, 60s)

> There's the trade-off isn't there? With quality of life. (P23, female, 50s)

While some felt so vulnerable that they considered it essential to be 'hypervigilant', 'so careful' and 'a little bit oversensitive' (P23, female, 50s), several participants expressed wariness of becoming, or being perceived as, a hypochondriac.

> There's a fine line between taking care and allowing your concerns to impact on your life. (P21, female, 60s)

### Changing experiences because of COVID-19

Participants' perceptions of RTIs and prevention behaviours were influenced by the context of the COVID-19 pandemic. Participants recognised COVID-19 as an RTI, although one that is unfamiliar and particularly severe, creating heightened motivation for RTI prevention.

> Particularly now, you need to be armed to the teeth with everything you can to protect yourself! (P23, female, 50s)

Several participants were fearful, perceiving themselves to be at risk of severe COVID-19 due to the same vulnerabilities that they understood predisposed them to other RTIs.

> I'm paranoid. Absolutely paranoid about getting it because I can't breathe very well anyway (P15, female, 50s)

Participants recognised similarities in transmission mechanisms between COVID-19 and non-pandemic RTIs and described developing better understandings about spread of infections and increasing insight into how to stop transmission, particularly hygiene measures.

> Before this, I wasn't so aware of how illnesses travel, especially a cough and a sneeze […] and how long it stays on the surface. (P14, female, <20)

> We're all used to using these hand sanitisers and we understand how they're protecting us. (P23, female, 50s)

> I've definitely learnt through coronavirus that I touch my face all the time. (P10, female, 60s)

However, their sense that their health was at the mercy of other people appeared heightened; participants were extremely conscious and critical of other people's behaviour.

> They're mixing with people and then still going out and about the shops like their normal day-to-day living, and that's the scary bit. They don't understand that they might not have it, but they might be a carrier, and carry it to people like me. (P8, female, 50s)

During national lockdown, shielding and other restrictions, participants' usual behaviour of distancing themselves from symptomatic people changed to distancing from almost everyone. Consequently, participants experienced an unequivocally successful RTI prevention method, often for the first time.

> Where you're not mixing with people, you're not catching colds! (P21, female, 60s)

Some participants found social distancing and isolation burdensome.

> I'm questioning where that trade-off is comfortable. (P23, female, 50s)

Nonetheless some had discovered approaches that they speculated adopting in the longer term for 'normal' RTIs.

> Being maybe more cautious in social interactions, to prevent catching things anyway. Maybe we'll all be like that! (P20, male, 50s)

## DISCUSSION
### Novel contributions and implications

This study extends the literature by exploring how a recurrent and at-risk patient group experiences and understands their RTIs, including if and how they can be prevented. Some findings echoed those from previous research but several important new insights emerged. These may be useful for determining how to successfully engage these patients in RTI prevention behaviours and are therefore discussed in these terms below.

Like previous studies on colds and influenza,[24–26 29 30] our participants demonstrated a largely accurate understanding of RTI transmission. Their identification of children as important infectors (for non-pandemic RTIs) came out particularly strongly in this study, compared with past research. Alongside their correct understandings of virus transmission, many participants held concurrent beliefs about the causal role of temperature, damp, allergens and pollution. Such beliefs have been noted previously, but sometimes considered erroneous, folklore or culture-specific.[31] However, these factors could be relevant; modest evidence suggests that cold and damp could be causally linked to RTIs.[43 44] Furthermore, it is widely accepted that respiratory viruses thrive in cold and damp environments. Another explanation is that patients may confuse a causal role of these factors with their tendency to trigger or exacerbate respiratory symptoms from their chronic conditions (eg, coughs and wheezes from asthma or allergies). Addressing causal beliefs may be necessary if an intervention relies on accurate understandings of virus transmission.

Unlike some earlier studies,[27 29] our patients were highly motivated to prevent RTIs and engaged in many relevant behaviours. Consistent with their grasp of transmission mechanisms, they focused (even pre-COVID-19) on avoiding contact with symptomatic others and undertaking hygiene measures. Previous literature, in contrast, suggested such measures might be considered unfeasible, unnecessary or unacceptable.[27 29 41] Our participants were open to prevention recommendations and described extending their prevention repertoire over time. Furthermore, some were considering continuing with COVID-19 measures such as social distancing and mask-wearing to avoid routine RTIs in the future. It therefore seems these recurrent and at-risk patients will be highly receptive to additional support with infection prevention. The differences in motivation and acceptance of prevention behaviours between the current study and previous research likely reflects our sampling of a higher risk/higher impact patient group but may also reflect the heightened concern due to the COVID-19 context. Contrary to previous research where RTIs have often been described as trivial and participants have not considered themselves susceptible to catching RTIs or vulnerable to severe consequences[24 26 27 29] most of our participants perceived a personal vulnerability to frequent and/or severe infections. Some described how RTIs were linked to medical conditions, stress was salient for others, some had vague ideas about weak immune systems, and other showed uncertainty and frustration about vulnerability. Interventions that offer an acceptable explanatory framework for recurrent infections, help people understand their personal vulnerability, and provide a corresponding, coherent prevention strategy may prove validating and helpful to these patients.

As in some previous studies,[40 41] some participants described engaging in behaviours to stay healthy and strong. However, our analysis highlighted how most of these efforts appeared to be engaged in through enjoyment, habit or in the service of managing other health conditions. Few participants appeared to have consciously considered and committed to boosting their immune system to prevent RTIs, though, for example, increased physical activity, adequate sleep and stress reduction. No participants mentioned recommendations from health professionals to make lifestyle changes for RTI prevention. Patients suffering from recurrent or severe RTIs should be given information about how these behaviours relate to immune health. They may benefit from support and encouragement with initiating, stepping up, and maintaining these behaviours.

Despite seeking to avoid RTIs, participants had some doubts about whether prevention is indeed possible or desirable and were unclear about the extent to which their strategies were helping them. Fatalism about RTIs has been identified previously[24 26 41] but, importantly, here participants describe helplessness despite considerable prevention efforts; the idea of being at the mercy of other people's risky and inconsiderate behaviour is apparent as is a sense of unfairness. Interventions will need to improve outcome expectancies by convincing people that prevention approaches will be effective, and by helping people to identify situations and behaviours where they can exert personal control, despite the role of other people in infection transmission. A novel finding was that many participants believed that they sometimes succeeded in stopping RTIs developing or worsening through rapid intervention when they begin. These anecdotes suggest people perceive their own actions sometimes alter the progression of RTIs. Drawing on these experiences might help people to develop confidence about other approaches to preventing infections. Finally, participants expressed concerns about prevention behaviours dominating life. Concerns about prevention strategies may require more detailed research, including worries or barriers specific to particular approaches. Acknowledging and addressing concerns and careful communication and framing may be necessary to make prevention interventions acceptable.

### Strengths and limitations

While our recruitment strategy successfully identified recurrent and at risk RTI patients, the sample was self-selecting and nesting our data collection within an RTI prevention intervention development study may have attracted participants who were particularly interested in prevention. This may limit the transferability of findings.

Furthermore, we are unlikely to have included the experiences of underserved populations. Although we recruited from practices serving a variety of communities and included six participants with index of multiple deprivation deciles <4, the sample was skewed towards low deprivation. Of those that provided ethnicity data, all but one were White British/Irish. This is a limitation, especially given the increasing awareness of the unequal susceptibility to and impact of COVID-19.[45–48] Capturing the perspectives of underserved populations who are also unequally burdened by the impact of RTIs remains a priority.

## Conclusion

Our participants described multifaceted causal understandings of their recurrent and/or severe RTIs, including both transmission from others and their own perceived weakened defences. Despite trying various strategies to prevent infections, RTIs persisted and they often described feeling powerless to influence infections. People with a history of recurrent RTIs seem likely to welcome prevention interventions but may need to be persuaded that approaches will be effective, drawing on a convincing and personally meaningful rationale for how interventions will help them. Our study shows how COVID-19 has facilitated learning about RTI spread, heightened feelings of vulnerability but also exposed participants to highly effective prevention strategies. The current heightened concern and awareness about infections may be fertile ground for introducing prevention approaches for non-pandemic infections, at least in populations who already perceive themselves to be badly affected or vulnerable. Future research should explore how people respond to interventions to help them reduce RTI frequency and severity.

**Acknowledgements** Kate Martinson manged ethical approvals and recruitment. Thanks also to our public contributors Hazel Patel, Debs Smith and Samantha Richards-Hall.

**Contributors** AWAG is the guarantor for this work. AWAG, LY and PL conceived the study idea and initial study design with later input in planning the study from KB, LD, KG, SW and JV. SW and LD collected the data. LD led data analysis with analytic and interpretive input from SW, KG, MH, KB, JV, LY and AWAG at different stages. LD drafted the manuscript and SW, KG, MH, KB, JV, LY, PL and AWAG contributed to critically editing and approving the final manuscript.

**Funding** This article presents independent research funded by the National Institute for Health Research (NIHR) under its Programme Grants for Applied Research (PGfAR) Programme (Grant Reference Number RP-PG-0218-20005). The views expressed are those of the authors and not necessarily those of the NIHR or the Department of Health and Social Care. LY is an NIHR Senior Investigator and her research programme is partly supported by NIHR Applied Research Collaboration (ARC)-West, NIHR Health Protection Research Unit (HPRU) for Behavioural Science and Evaluation and the NIHR Southampton Biomedical Research Centre (BRC).

**Competing interests** None declared.

**Patient and public involvement** Patients and/or the public were involved in the design, or conduct, or reporting, or dissemination plans of this research. Refer to the Methods section for further details.

**Patient consent for publication** Not applicable.

**Ethics approval** This study involves human participants. Ethics and research governance approvals were granted by NHS and the University of Southampton (REC/HRA19/SC/0354; ERGO:48223). Participants provided written or online consent after reading the participant information sheet and having the opportunity to discuss participation with the research team. They were free to withdraw at any time. Procedures were in place to pause or discontinue interviews should participants become distressed, however this was not necessary. Participants were assured of confidentiality and quotations from their interviews have been anonymised. Participants were debriefed and received a £10 retail voucher to thank them for their time.

**Provenance and peer review** Not commissioned; externally peer reviewed.

**Data availability statement** Data are available upon reasonable request. The data that support the findings of this study are available from the corresponding author upon reasonable request.

**ORCID iDs**
Laura Dennison http://orcid.org/0000-0003-0122-6610
Sian Williamson http://orcid.org/0000-0001-5448-3499
Kate Greenwell http://orcid.org/0000-0002-3662-1488
Jane Vennik http://orcid.org/0000-0003-4602-9805
Paul Little http://orcid.org/0000-0003-3664-1873
Adam W A Geraghty http://orcid.org/0000-0001-7984-8351

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
