## [Reviewer comments · BMJ Open]

ARTICLE DETAILS

TITLE (PROVISIONAL)	Patient perceptions of vulnerability to recurrent respiratory tract infections and prevention strategies; a qualitative study
AUTHORS	Dennison, Laura; Williamson, Sian; Greenwell, Kate; Handcock, Molly; Bradbury, Katherine; Vennik, Jane; Yardley, Lucy; Little, Paul; Geraghty, Adam

VERSION 1 – REVIEW

REVIEWER	Lindberg, Bent H. Hamar Out Of Hours Primary Care Ctr
REVIEW RETURNED	21-Oct-2021

GENERAL COMMENTS	Thank you for this interesting and pertinent study on an important topic. Some reflexivity first: I am a GP, specialist in Family Medicine. My field of research is respiratory tract infections and antibiotic prescribing, and my qualitative research background is within the tradition of systematic text condensation. This is also a thematic, inductive method, aiming to let themes emerge from the text material. I have some general comments to the manuscript: The research question seems well defined in the abstract, page three, line 15-18. However, I do not find a clear research question in the introduction, and there is no objective described either. The lines 39-57 at page six describes some objectives with the study, but it appears somewhat unclear and not well defined. "We originally aimed to explore..." should be followed by something that you actually explored and aimed for. The interviews took place during the COVID-19 pandemic, and because of this only two of 23 persons were interviewed face-to-face. When I read the manuscript, it is as if though I sense the presence of the pandemic through the whole section of findings. However, COVID-19 is only mentioned in one of four main themes. I find this a bit odd. Would you consider rewriting, or explaining better how the other three themes are excluded from the influence of COVID-19? To me, the citations that are intermingled in the descriptive text (example on p 13, line 46-59) are confusing, making the distinction between interpretations and citations unclear. Would you consider rewriting these paragraphs, drawing all citations out of the main text? Some findings are supported by two or three citations, example on p
--

	15. Do these citations really add value to the analysis? In my opinion it may leave an impression of a lack of in-depth analysis when you add so many citations to a small passage of interpretations. Page 5, line 34-41. The conclusion on the efficacy of different approaches is unclear. Maybe the conclusion is unnecessary? Or would you consider specifying which approaches that are well-established? Page 6, line 10-16. Long sentence, a bit difficult to grasp. Consider splitting. Page 7, line 33-35. How would RTIs be recorded in medical records without consultations? Page 7, line 38. How or why was this practice chosen? Page 8, line 43-47. The reflexivity about who did what in this study could have been clearer. What is input? Did all the six authors engage in the iterative process? Table 2: RTIs per 3 years, and last 12 months; reported by whom? Informants or GPs? Page 14, line 41. "A minority" – to me this term gives a statistical impression. Would you rephrase? Page 17, the sub theme "Wondering why" is (in my opinion) one of the most interesting findings. Would you consider elaborating? Page 26, line 58 and out. This conclusion is probably right, but I find it too little enlightened, ref what I have mentioned above.
--	--

REVIEWER	González-Olmo, María José Rey Juan Carlos University
REVIEW RETURNED	05-Nov-2021

GENERAL COMMENTS	I find the article interesting. Here are some comments that could be useful for the revision of the article Add reference on the importance of oral hygiene in the transmission of COVID-19 (González-Olmo MJ, Delgado-Ramos B, Ruiz-Guillén A, Romero-Maroto M, Carrillo-Díaz M. Oral hygiene habits and possible transmission of COVID-19 among cohabitants. BMC Oral Health. 2020 Oct 19;20(1):286. doi: 10.1186/s12903-020-01274-5. PMID: 33076880; PMCID: PMC7569355.) Page 5, line 37 please add in the discussion section the limitation derived from the use of qualitative analysis of the variables.
--

REVIEWER	Hayes, Catherine Public Health England
REVIEW RETURNED	05-Jan-2022

GENERAL COMMENTS	This paper covers a qualitative investigation of the perceptions of patients on recurrent respiratory tract infections. This is a topic of clear importance and the authors have thoughtfully placed their study within the context of the literature and identified a clear evidence need. The methodology is sound and the results are presently clearly with novel findings which can help us understand
--

how to better support patients with recurrent RTIs. This work was planned before the COVID-19 pandemic but the authors adapted their methodology accordingly and have considered the impact and discuss this within the introduction and discussion sections. There are some limitations in terms of the diversity of participants which the authors have recognised.

Please consider the below questions and comments to further enhance the clarity of the paper:

1. There are currently no results shared about the intervention - are these reported elsewhere? I suggest you include these results or reduce the detail about the second sections of the interviews (e.g. in table 1) and just explain for context that interviews also involved discussion of an intervention, but this is not presented in the paper. If you keep the references to the intervention, please can the link to the intervention be included, or images/ intervention functions?

2. Naturally, some interviews had to be done remotely during the COVID-19 pandemic. How, if at all did this impact your engagement and recruitment of participants? Could there have been any barriers to participant through this method. Could there be differences in the findings collected through telephone and in person, for instance not being able to read body language, or building rapport as effectively.

3. Were there any ethical considerations to participating and how did researchers address these?

4. PPI group - please give more details about the panel, did any of the representative have experience of recurrent RTIs?

5. Recruitment - you considered that there may be many patients with recurrent RTIs who don't consult, please discuss your choice to recruit through general practice and was there a reason participants could not have been recruited from the community?

6. Data analysis - was there a double coder or did one researcher conduct the thematic analysis? This may lead to biases in the analysis and should be discussed by researchers in the discussion.

7. Please provide more information about the researchers and how they engaged in reflexivity? In your COREQ checklist you state interviewer characteristics are covered in the Method, Discussion, Funding statement sections. I cannot find the information in the discussion, and the information in the funding and methods do not consider assumptions and how these were mitigated. Please explain if there is any researcher bias in finding the emergent themes, especially as one author conducted the thematic analysis.

8. Data saturation and stopping interviews - the statement currently included in the COREQ checklist around ceasing recruitment once intervention development was concluded should be included in the manuscript.

9. Were there any differences in emerging themes according to participant demographics? e.g. Health literacy, deprivation. You have identified an imbalance of participants by gender please discuss any differences, and if the findings may have been different if there was an equal balance in gender.

	Other minor:  1. Page 9 line 31 is confusing, are the brackets saying the number they interviewed? Suggest presenting this more clearly. 2. Page 9, line 34 - was this timing just section 1 of interview or inclusive of discussion of the intervention? 3. Table 2 Figures don't line up under employment status
--	--

VERSION 1 – AUTHOR RESPONSE

REVIEWER 1

Thank you to Dr Lindberg for the encouraging comments about the importance of this work. We present responses to their detailed and constructive suggestions below.

- 1. The research question seems well defined in the abstract, page three, line 15-18. However, I do not find a clear research question in the introduction, and there is no objective described either. The lines 39-57 at page six describes some objectives with the study, but it appears somewhat unclear and not well defined. “We originally aimed to explore...” should be followed by something that you actually explored and aimed for.**

Our phrasing at the end of the introduction was a little unclear as we did indeed explore what we aimed to, despite the unanticipated pandemic changing the context. We have changed the wording so that the research question is phrased in the same way as in the abstract and the wording ‘originally’ has been removed. There was not a deviation from our original research question but there was an intensification of the need to conduct the research as a consequence of the pandemic, which the final paragraph of the introduction explains. Our research questions were exploratory and broad – around how people understand RTIs, their vulnerability and infection prevention behaviours (already in the text). We do not have further specific or well-defined objectives except that we intend to use the findings to understand how to support and engage people with RTI prevention behaviours (already in the text).

- 2. The interviews took place during the COVID-19 pandemic, and because of this only two of 23 persons were interviewed face-to-face. When I read the manuscript, it is as if though I sense the presence of the pandemic through the whole section of findings. However, COVID-19 is only mentioned in one of four main themes. I find this a bit odd. Would you consider rewriting, or explaining better how the other three themes are excluded from the influence of COVID-19?**

At the time of interviews (spring and summer 2020) participants tended to view COVID-19 as very new, unfamiliar, threatening, and different to their normal RTIs and so most of the interviews consisted of in-depth discussions of their own typical pre-pandemic RTIs, with which they were extremely familiar and about which they had clear beliefs and engrained patterns of behaviour. Although there was some discussion of COVID-19 there was less than we as researchers expected. Themes 1,2 and 3 were developed around this talk about familiar infections. However, these themes are certainly not excluded from the influence of COVID-19. We like the way Reviewer 1 refers to ‘sensing the presence of the pandemic’ – it does indeed feel like it exerts influence throughout, in various ways. We have thought about how to get that across, whilst preserving the participants’ focus on the familiar, seasonal viral illnesses. We now refer to theme 4 (*Changing Experiences because of COVID-19*) as an overarching theme, rather than separate (see changes to start of Findings section and table 2).

- 3. To me, the citations that are intermingled in the descriptive text (example on p 13, line 46-59) are confusing, making the distinction between interpretations and citations unclear. Would you consider rewriting these paragraphs, drawing all citations out of the main text?**

By 'citations' I believe the reviewer meant quotations. Most of these have now been re-formatted/rewritten except in a handful of cases where the quotations were so short that they did read better within the descriptive text

- 4. Some findings are supported by two or three citations, example on p 15. Do these citations really add value to the analysis? In my opinion it may leave an impression of a lack of in-depth analysis when you add so many citations to a small passage of interpretations.**

We have cut out 11 quotations. We have also moved most of the short quotations previously intermingled in the descriptive text (as above). This should help the author's description and discussion of themes and subthemes to stand out more from the quotations. We retain most of the quotations as these improve transparency in the presentation of the analysis. They allow readers to judge our interpretations against raw data and vividly portray some of the variety and nuance within the themes and subthemes. In comparison with other qualitative studies published in BMJ Open and elsewhere the ratio of quotations to theme discussions does not appear unusual.

- 5. Page 5, line 34-41. The conclusion on the efficacy of different approaches is unclear. Maybe the conclusion is unnecessary? Or would you consider specifying which approaches that are well established?**

We agree this is an unnecessary point to make and have removed the sentence.

- 6. Page 6, line 10-16. Long sentence, a bit difficult to grasp. Consider splitting.**

Done.

- 7. Page 7, line 33-35. How would RTIs be recorded in medical records without consultations?**

Even though not all RTIs are recorded in medical records (e.g. many colds and other common RTIs are self-managed without consultation), some infections are consulted for. This is particularly true of patients who have severe infections and those are prescribed antibiotics. We therefore devised a search strategy to identify patients who were recorded as seeking healthcare for RTIs and also captured patients with co-morbid chronic conditions that are relevant to severe and/or recurrent infections. This method was likely to target patients for whom RTIs are particularly troublesome. Because we know that medical records do not record every RTI, in addition to identifying RTI consultations for selecting patients to invite we needed prospective participants self-report numbers of RTIs over the last 12 months to clarify whether they were indeed suffering from recurrent RTIs.

Please note that this somewhat complicated recruitment method (compared to the community recruitment suggested by Reviewer 3, comment #5) was related to an upcoming RCT. We were piloting our search and mailout recruitment strategy.

- 8. Page 7, line 38. How or why was this practice chosen?**

Clarified. GP practices were chosen to achieve a both urban and semi-rural locations and varying deprivation levels.

- 9. Page 8, line 43-47. The reflexivity about who did what in this study could have been clearer. What is input? Did all the six authors engage in the iterative process?**

We've clarified more about the roles and contributions of the authors involved in analysis. LD conducted the thematic analysis. Co-authors were involved in discussions about codes and early themes and subthemes to allow multiple perspectives to be considered, avoid idiosyncratic interpretations, highlight modifications or clarifications to improve the analysis, and to support the main analyst in refining and communicating the analysis.

(See also our response to a similar point from reviewer 3 – comment #6)

10. Table 2: RTIs per 3 years, and last 12 months; reported by whom? Informants or GPs?

Edited to show this was self-report.

11. Page 14, line 41. "A minority" – to me this term gives a statistical impression. Would you rephrase?

Rephrased to 'a few', maintaining the sense that this was present but not common in our sample.

12. Page 17, the sub theme "Wondering why" is (in my opinion) one of the most interesting findings. Would you consider elaborating?

Yes, we also found this interesting, especially given the participants ability to identify how viral transmission works and reflect on possible personal factors that may underlie their vulnerability. The idea that being vulnerable was unfair stood out. As requested we have added a little more detail and analytic insight to this paragraph.

13. Page 26, line 58 and out. This conclusion is probably right, but I find it too little enlightened, ref what I have mentioned above.

We have amended this conclusion a little, adding more caution and tentativeness.

REVIEWER 2

Thank you to Dr González-Olmo for her time spent considering the article and her comments about finding it of interest. We respond to her suggestions below.

- 1. Add reference on the importance of oral hygiene in the transmission of COVID-19 (González-Olmo MJ, Delgado-Ramos B, Ruiz-Guillén A, Romero-Maroto M, Carrillo-Díaz M. Oral hygiene habits and possible transmission of COVID-19 among cohabitants. BMC Oral Health. 2020 Oct 19;20(1):286. doi: 10.1186/s12903-020-01274-5. PMID: 33076880; PMCID: PMC7569355.) Page 5, line 37**

Thank you for alerting us to this article. This was a very interesting read but we do not feel it is sufficiently relevant to add to the introduction of the current paper (See Editor's comment).

- 2. Please add in the discussion section the limitation derived from the use of qualitative analysis of the variables.**

We appreciate this point, however, as a qualitative study there are no pre-determined 'variables' to analyse; instead inductive themes were developed through rigorous analysis of the data. We do not feel it is relevant or helpful to discuss using qualitative analysis as a limitation of a qualitative research design any more than a quantitative study would draw attention to overall limitations of statistical

analysis. Our limitations section already draws out some *specific* limitations of our qualitative methods (now including some additional issues, raised by other reviewers such as more discussion of bias and reflexivity).

REVIEWER 3

Thank you to Dr Hayes for her encouraging comments about the importance of the research, its sound methodology, and novel and important findings. We present responses to her very thorough review and helpful suggestions below.

1. There are currently no results shared about the intervention - are these reported elsewhere? I suggest you include these results or reduce the detail about the second sections of the interviews (e.g. in table 1) and just explain for context that interviews also involved discussion of an intervention, but this is not presented in the paper. If you keep the references to the intervention, please can the link to the intervention be included, or images/ intervention functions?

We now explicitly state that the intervention development and refinement will be published elsewhere (it is currently under review so we can not cite it yet). As suggested, we have reduced the amount of information about the second section of the interview. Section 1 and section 2 are now only described in the main text, and the table has been removed, improving conciseness of the section overall.

A description of section 2 does need to be included because data from this section did contribute to the current analysis. Even though section 2 focussed on participants viewing/trying interventions and providing feedback, participants frequently described more about their RTI experiences, beliefs and behaviours during this section. We therefore analysed the entire interview, drawing on any data that helped answer our research questions. For example, when introduced to the nasal spray intervention some participants described their understandings of viral transmission and when introduced to the physical activity or intervention participants would often spontaneously discuss their own lifestyle and health behaviours and how they feel it relate to their vulnerability to RTIs and ability to fight them off.

2. Naturally, some interviews had to be done remotely during the COVID-19 pandemic. How, if at all did this impact your engagement and recruitment of participants? Could there have been any barriers to participant through this method. Could there be differences in the findings collected through telephone and in person, for instance not being able to read body language, or building rapport as effectively.

We can't really draw any scientific conclusions on this, only hunches. We had no difficulty recruiting participants (perhaps the beginning of the pandemic made the study seem interesting, important and relevant?). Interviews went smoothly with participants talking in detail and at length about their RTI beliefs and behaviours and very few problems understanding and meaningfully connecting with each other. The barriers mentioned are all possible and certainly body language is notably absent. However, our experience within this study and many other studies is that skilled/experienced interviewers (which we had) are able to quickly and effectively build rapport over the telephone and facilitate in depth, sensitive discussions. Some of the barriers may even be outweighed by easier recruitment due to greater convenience, anonymity and the ability to reach people who otherwise would not participate. As we can only speculate on whether the telephone interviewing improved or hindered recruitment or data collection, and had no choice but to adopt telephone interviewing during the first wave of COVID-19, we propose not to add a discussion of this to the text of the article itself.

3. Were there any ethical considerations to participating and how did researchers address these?

We now outline the key issues in the ethical approval section of the method (some of which we moved from other sections so that it is all together).

4. PPI group - please give more details about the panel, did any of the representative have experience of recurrent RTIs?

We now provide more information about the PPI panel members. They have experience of recurrent RTIs and/or long term health conditions that put them at risk of more severe RTIs

5. Recruitment - you considered that there may be many patients with recurrent RTIs who don't consult, please discuss your choice to recruit through general practice and was there a reason participants could not have been recruited from the community?

It would have been possible to find participants via community sampling. However, as a study funded via an NIHR Programme Grant, we recruited via the NHS. Our choice to use search and mailout from GP practices was influenced by

- a) The need to efficiently identify and invite people with health conditions of importance for recurrent and severe RTIs (specifically we looked for COPD, Asthma and Chronic sinusitis) and to try to sample some of the recurrent RTI patients who may be overusing antibiotics.
- b) The need to pilot recruitment methods and estimate response ahead of our upcoming RCT of infection prevention approaches (which was due to target the recurrent and/or at risk population).

We feel this justification would make the recruitment section overly lengthy but will consider adding to the main article if the reviewer and editor feel this is essential to explain.

6. Data analysis - was there a double coder or did one researcher conduct the thematic analysis? This may lead to biases in the analysis and should be discussed by researchers in the discussion.

We have clarified the roles and contributions of the authors involved in analysis (page 8). LD conducted the thematic analysis. Co-authors were involved in discussions about codes and early themes and subthemes to allow multiple perspectives to be considered, avoid idiosyncratic interpretations, highlight modifications or clarifications to improve the analysis, and to support the main analyst in refining and communicating the analysis.

There was no double coding. We do not seek or claim an objective and unbiased analysis and disagree that an unbiased analysis would be achieved with double coding. Rather, we present a rigorous, transparent qualitative study, achieving quality criteria appropriate to qualitative research methods. We have discussed this in the analysis section (page 8).

7. Please provide more information about the researchers and how they engaged in reflexivity? In your COREQ checklist you state interviewer characteristics are covered in the Method, Discussion, Funding statement sections. I cannot find the information in the discussion, and the information in the funding and methods do not consider assumptions and how these were mitigated. Please explain if there is any researcher bias in finding the emergent themes, especially as one author conducted the thematic analysis.

We have removed reference to 'discussion' from the COREQ checklist.

We have developed the analysis section to discuss more about the backgrounds of the researchers and their influence. This relates to the point above about bias- we do not believe it can be removed, or that double coding would remove bias.

8. Data saturation and stopping interviews - the statement currently included in the COREQ checklist around ceasing recruitment once intervention development was concluded should be included in the manuscript.

Added to recruitment section of method.

9. Were there any differences in emerging themes according to participant demographics? e.g. Health literacy, deprivation. You have identified an imbalance of participants by gender please discuss any differences, and if the findings may have been different if there was an equal balance in gender.

The following sentence has been added the beginning of the findings section “Unless described, no clear patterns relating to demographic characteristics were discerned”.

The qualitative research design and sample size of 23 prevented any robust comparisons or conclusions about demographic differences. We did not develop any hunches about trends relating to gender and we cannot speculate on how findings may have been different with more men in the sample. The only finding that clearly stood out as relevant to some age groups more than others related to the perceived impact of ageing, stress and post-retirement lifestyle, and being in contact with children as a parent or grandparent (already discussed in *Defective bodies and inadequate defences* subtheme and *Catching RTIs from other people* subtheme p 12).

Other minor:

1. Page 9 line 31 is confusing, are the brackets saying the number they interviewed? Suggest presenting this more clearly.

We have rewritten this more clearly.

2. Page 9, line 34 - was this timing just section 1 of interview or inclusive of discussion of the intervention?

We have clarified that this reflects the whole interview, including discussion of the intervention.

3. Table 2 Figures don't line up under employment status

Now aligned.

EDITORIAL COMMENTS:

During our internal evaluation, we noted that reviewer #2 has requested for a reference to be cited in the manuscript. Please note that addressing this request is not mandatory but at the authors' discretion.

Noted. We have decided that although it is an interesting paper, this reference is not relevant enough to include.

EDITORIAL REQUESTS:

- Please include page numbers on your checklist, indicating where each item can be found in your manuscript.

Added.

FORMATTING AMENDMENTS (WHERE APPLICABLE):

- Please ensure the ethics approval statement in main document file is under the heading 'Ethics Approval'.

It is already in this location. I believe this was picked up and changed soon after our original submission and before peer review.

FURTHER NOTES FROM AUTHORS ABOUT MINOR ADDITIONS/EDITS

1) Since submitting the paper in summer 2021 we have been granted approval from our ethics committee to return to participants to ask their ethnicity. We have completed this retrospective data

collection (17/23 responded and provided data). We therefore now present ethnicity data in table 2 and have edited our discussion of ethnic diversity in the discussion (p25)

2) We have become aware of guidance to avoid giving multiple demographic characteristics of participants alongside quotes to avoid inadvertently making them more identifiable. We have therefore change precise ages to age ranges.

3) We have also taken the opportunity to correct the standardised wording about our funding source used in our funding statements.

VERSION 2 – REVIEW

REVIEWER	Lindberg, Bent H. Hamar Out Of Hours Primary Care Ctr
REVIEW RETURNED	08-Feb-2022

GENERAL COMMENTS	Thank you for your revised manuscript, which I find improved, interesting and well written. I have only a few minor comments: Line 9 in the abstract: Should the word "prescriptions" be "prescribing?" Line 19 on page 7: "Procedures were in place..." is a bit difficult to understand, due to the lack of a comma? Line 38 on page 8: To me this seems obvious with a qualitative study, being more of a response to criticism from a reviewer than a necessary phrase in the manuscript. However, I leave that up to the authors and/or the editor to decide.
--

REVIEWER	Hayes, Catherine Public Health England
REVIEW RETURNED	04-Feb-2022

GENERAL COMMENTS	Thank you for this interesting study and for an excellent revision. This is a crucial topic with novel findings for patients with recurrent RTIs. Best of luck with publication.
---

VERSION 2 – AUTHOR RESPONSE

Reviewer: 3

Dr. Catherine Hayes, Public Health England Comments to the Author:

Thank you for this interesting study and for an excellent revision. This is a crucial topic with novel findings for patients with recurrent RTIs. Best of luck with publication.

Thank you for your encouraging comments and thorough review.

Reviewer: 1

Dr. Bent H. Lindberg, Hamar Out Of Hours Primary Care Ctr Comments to the Author:

Dear authors.

Thank you for your revised manuscript, which I find improved, interesting and well written. I have only a few minor comments:

Thank you for your encouraging comments and thorough review.

Line 9 in the abstract: Should the word "prescriptions" be "prescribing?"

Yes. Edited.

Line 19 on page 7: "Procedures were in place..." is a bit difficult to understand, due to the lack of a comma?

Comma added.

Line 38 on page 8: To me this seems obvious with a qualitative study, being more of a response to criticism from a reviewer than a necessary phrase in the manuscript. However, I leave that up to the authors and/or the editor to decide.

Note to editor: Reviewer 1 refers to the sentence "Unless described, no clear patterns in findings relating to demographic characteristics were discerned." This was indeed added in response to a comment from Reviewer 3. I agree it is obvious and have removed it in the current resubmission. However, should you prefer that it stays we do not object. I have pasted the original comments from reviewer 3 and responses from the authors below should you wish to consider this further.

Reviewer 3: Were there any differences in emerging themes according to participant demographics? e.g. Health literacy, deprivation. You have identified an imbalance of participants by gender please discuss any differences, and if the findings may have been different if there was an equal balance in gender.

Authors: The following sentence has been added the beginning of the findings section "Unless described, no clear patterns relating to demographic characteristics were discerned".

The qualitative research design and sample size of 23 prevented any robust comparisons or conclusions about demographic differences. We did not develop any hunches about trends relating to gender and we cannot speculate on how findings may have been different with more men in the sample. The only finding that clearly stood out as relevant to some age groups more than others related to the perceived impact of ageing, stress and post-retirement lifestyle, and being in contact with children as a parent or grandparent (already discussed in *Defective bodies and inadequate defences* subtheme and *Catching RTIs from other people* subtheme p 12).